# Asymmetric recognition of HIV-1 Envelope trimer by V1V2 loop-targeting antibodies

Haoqing Wang[1], Harry B Gristick[1], Louise Scharf[1†], Anthony P West Jr[1], Rachel P Galimidi[1], Michael S Seaman[2], Natalia T Freund[3‡], Michel C Nussenzweig[3], Pamela J Bjorkman[1*]

[1]Division of Biology and Biological Engineering, California Institute of Technology, Pasadena, United States; [2]Beth Israel Deaconess Medical Center, Boston, United States; [3]Laboratory of Molecular Immunology, The Rockefeller University, New York, United States

*For correspondence: bjorkman@caltech.edu

Present address: †23andMe, Mountain View, United States; ‡Department of Clinical Microbiology and Immunology, Sackler School of Medicine, Tel Aviv University, Tel Aviv, Israel

**Abstract** The HIV-1 envelope (Env) glycoprotein binds to host cell receptors to mediate membrane fusion. The prefusion Env trimer is stabilized by V1V2 loops that interact at the trimer apex. Broadly neutralizing antibodies (bNAbs) against V1V2 loops, exemplified by PG9, bind asymmetrically as a single Fab to the apex of the symmetric Env trimer using a protruding CDRH3 to penetrate the Env glycan shield. Here we characterized a distinct mode of V1V2 epitope recognition by the new bNAb BG1 in which two Fabs bind asymmetrically per Env trimer using a compact CDRH3. Comparisons between cryo-EM structures of Env trimer complexed with BG1 (6.2 Å resolution) and PG9 (11.5 Å resolution) revealed a new V1V2-targeting strategy by BG1. Analyses of the EM structures provided information relevant to vaccine design including molecular details for different modes of asymmetric recognition of Env trimer and a binding model for BG1 recognition of V1V2 involving glycan flexibility.

## Introduction

The HIV-1 envelope (Env) glycoprotein trimer, a trimer of gp120-gp41 heterodimers, is the infectious machinery that targets host cell receptors CD4 and CCR5/CXCR4 to mediate fusion between the viral and host membranes (*Wyatt and Sodroski, 1998*). As the only viral protein on the surface of HIV-1 virions, Env trimer is the sole target of neutralizing antibodies that prevent infection of host cells (*McCoy and Burton, 2017*).

Although an infected host produces antibodies against HIV-1 Env, most are non-neutralizing or exhibit strain-specific neutralization that is ineffective in the face of HIV's ability to create resistant variants through rapid mutation (*McCoy and Burton, 2017*). Only a subset of HIV-1–infected individuals produce potent and broadly neutralizing antibodies (bNAbs) (*Klein et al., 2013*; *Landais et al., 2016*), which can protect against and suppress infection in animal models (*Klein et al., 2012*; *Barouch et al., 2013*; *Shingai et al., 2013*) and exhibit antiviral activity in human clinical trials (*Caskey et al., 2015*, *2017*; *Lynch et al., 2015*). HIV-1 bNAbs differ from most other antibodies in several ways. First, unlike most humoral immune responses, bNAbs usually take 1–3 years to develop (*Klein et al., 2013*; *Landais et al., 2016*). Second, nearly all HIV-1 bNAbs exhibit high levels of somatic mutation, and many of these antibodies include long heavy chain complementarity determining region 3 (CDRH3) loops (*West et al., 2014*). These unusual features appear to be a significant barrier to eliciting such antibodies by immunization.

HIV-1 bNAbs have been mapped to distinct epitopes on HIV-1 Env including the variable 1 and 2 (V1V2) and variable 3 (V3) loops, the CD4 binding site (CD4bs) on gp120, the gp120-gp41 interface, and regions of gp41 including the membrane-proximal external region and fusion peptide (*McCoy and Burton, 2017*; *Landais et al., 2016*; *West et al., 2014*; *Burton and Mascola, 2015*). The V1V2 epitope is of interest because this region of gp120 undergoes large structural changes after binding to the primary receptor CD4: the interaction between CD4 and the gp120 portion of Env trimer stabilizes an open Env conformation that can interact with a chemokine receptor to induce further conformational changes resulting in insertion of the fusion peptide on gp41 into the target cell membrane (*Wyatt and Sodroski, 1998*). Structures of soluble native-like Env trimers in the closed, prefusion state (*Garces et al., 2015*; *Gristick et al., 2016*; *Julien et al., 2013a*; *Kong et al., 2015*; *Kwon et al., 2015*; *Lyumkis et al., 2013*; *Pancera et al., 2014*; *Scharf et al., 2015*; *Stewart-Jones et al., 2016*; *Ward and Wilson, 2017*) showed that the V1V2 regions from the three gp120s interact around the trimer axis of symmetry to form a top layer of the trimer that shields V3 and the coreceptor binding site. In the CD4-bound state, the gp120 protomers are separated from each other (*Liu et al., 2008*; *Tran et al., 2012*), and the V1V2 loops are displaced by ~40 Å from the trimer apex to interact with CD4 on the sides of the trimer (*Wang et al., 2016*).

Structural studies of V1V2 bNAbs include crystal structures of bNAb Fabs bound to monomeric V1V2 scaffolds (*Gorman et al., 2016*; *McLellan et al., 2011*; *Pancera et al., 2013*; *Pan et al., 2015*) and a low resolution single-particle EM structure of the V1V2 bNAb PG9 bound to a soluble, native-like Env trimer (*Julien et al., 2013b*). Crystal structures of monomeric V1V2 scaffolds bound to PG9 or to its PG16 relative revealed the V1V2 loop structure as a Greek key motif containing four $\beta$-strands that was recognized by the V1V2 bNAbs largely through interactions in which a long CDRH3 from the Fab reached through the Env glycan shield to contact a protein epitope in the V2 loop (*McLellan et al., 2011*; *Pancera et al., 2013*). The EM structure and complementary biophysical studies indicated that, unlike bNAbs in other epitope classes that bind symmetrically to Env trimers with three Fabs per trimer, only one PG9 bound per Env trimer, resulting in an asymmetric quaternary epitope at the Env apex (*Julien et al., 2013b*). Recent cryo-EM structures of Env trimer complexes with the V1V2 bNAb PGT145 also revealed one V1V2 Fab per Env trimer and details of quaternary contacts spanning the three gp120 protomers of Env (*Liu et al., 2017*; *Lee et al., 2017*).

A new V1V2 bNAb, BG1, was isolated along with bNAbs targeting non-overlapping sites on Env from an HIV-1 controller who developed elite levels of HIV-1 neutralizing activity (*Freund et al., 2017*). We characterized the binding properties of BG1 by performing biophysical studies to determine its stoichiometry of binding to an HIV-1 Env trimer and by solving structures of the BG1 Fab and a BG1 Fab-Env trimer complex. BG1 differs from other V1V2 antibodies in that two BG1 Fabs bind per Env trimer to create a previously-unseen form of asymmetric recognition of the V1V2 loops at the Env trimer apex. This form of recognition does not require a protruding CDRH3, as found in other V1V2 bNAbs, and the BG1 epitope on Env trimer differs from previously-characterized V1V2 epitopes. The BG1 paratope may therefore represent a new target for HIV-1 vaccine development (*Escolano et al., 2017*).

## Results

### Crystal structure of BG1 Fab

The heavy chains (HCs) of V1V2-directed bNAbs isolated from multiple donors contain long (28–35 residues) anionic CDRH3 loops (*Figure 1A*) that include sulfated/sulfonated tyrosines (*McLellan et al., 2011*; *Pancera et al., 2013*; *Pan et al., 2015*; *Walker et al., 2011*, *2009*; *Bonsignori et al., 2011*; *Doria-Rose et al., 2014*). Structural studies of V1V2 bNAbs showed that the protruding CDRH3 is used to penetrate through the glycan shield on Env surrounding the $Asn160_{gp120}$ and $Asn156_{gp120}$ N-glycans to contact basic residues in the V2 loop (*Gorman et al., 2016*; *McLellan et al., 2011*; *Pancera et al., 2013*; *Pan et al., 2015*; *Pancera et al., 2010*; *Pejchal et al., 2010*). The BG1 CDRH3 length, 22 residues, is shorter than CDRH3 loops in other V1V2 bNAbs (*Figure 1A*), but longer than typical human antibody CDRH3s for which the most common length is 14 residues (*Swindells et al., 2017*). A 2.0 Å crystal structure of unbound BG1 Fab (*Figure 1B*; *Table 1*) revealed that, in contrast to the protruding CDRH3s in other V1V2 bNAbs, the BG1 CDRH3 does not extend notably beyond the antigen-binding site surface (*Figure 1C*). The

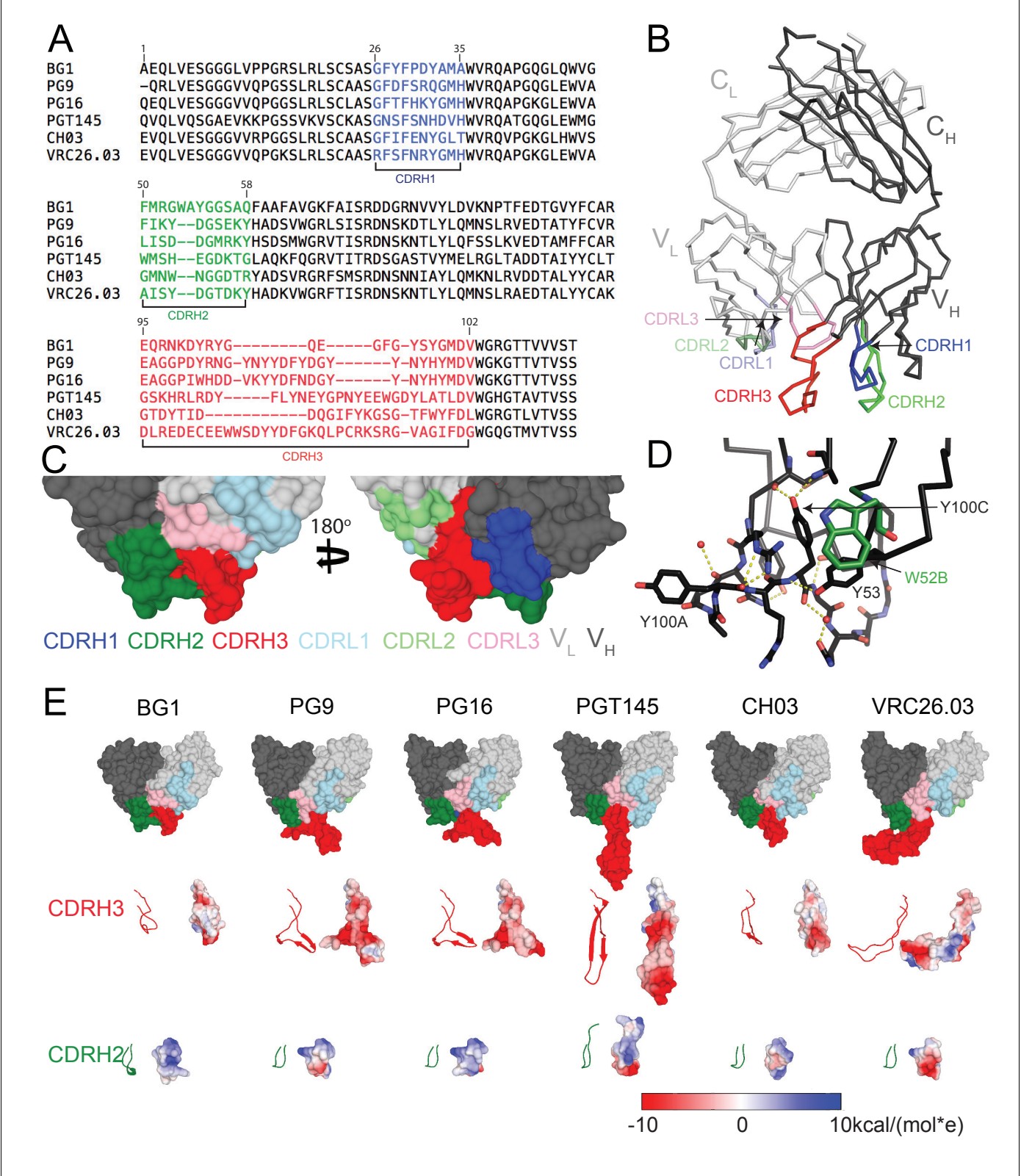

**Figure 1.** Comparison of BG1 and other V1V2 bNAbs. (**A**) Sequences of the HCs of BG1 and representative other V1V2 bNAbs. Residue numbering (Kabat) refers to BG1, and the CDRs were defined using AbM (*Swindells et al., 2017*). (**B**) Crystal structure of BG1 Fab with highlighted CDRs. (**C**) Space-filling representations of antigen-binding site of BG1 in two orientations with CDRs highlighted. (**D**) Hydrogen bonds (dotted yellow lines) contributing to compact structure of CDRH3. Water molecules shown as red spheres. (**E**) Top: Space-filling representations of $V_H$-$V_L$ domains of

*Figure 1 continued on next page*

eLIFE Research article                                                    Biophysics and Structural Biology

*Figure 1 continued*

selected V1V2 bNAbs (BG1: this study; PG9: PDB 3U4E; PG16: PDB 3 MUG; PGT145: PDB 3U1S; CH03: PDB 5ESV; VRC26.03: PDB 4OD1). Residues within CDRs are highlighted. CDRH3 (middle) and CDRH2 (bottom) loops shown in ribbon (left) and space-filling (right) representations. Electrostatic potentials are shown on the space filling representations using blue and red for positive and negative electrostatic potentials, respectively.

The following figure supplement is available for figure 1:

**Figure supplement 1.** Electron densities for tyrosines within BG1 Fab CDRs.

relatively compact CDRH3 of BG1 results from folding into an irregular structure whose conformation is stabilized by $Tyr100C_{CDRH3}$ interactions with protein backbone atoms in CDRH1 and CDRH3 and by water-mediated H-bonds involving other residues (*Figure 1D*). None of the four tyrosines within the BG1 CDRH3 ($Tyr100A_{CDRH3}$, $Tyr100C_{CDRH3}$, $Tyr100J_{CDRH3}$ and $Tyr100L_{CDRH3}$) or within

**Table 1.** BG1 Fab crystal structure data collection and refinement statistics.

|  | 354BG1 |
| --- | --- |
| **Data collection** | |
| Beamline | SSRL 12–2 |
| Wavelength | 0.9795 Å |
| Temperature (°K) | 100 |
| Space group | $P3_221$ |
| Cell dimensions | |
| $a, b, c$ (Å) | 95.6, 95.6, 103.9 |
| $\alpha, \beta, \gamma$ (°) | 90, 90, 120 |
| Resolution (Å) | 82.82–2.00 (2.05–2.00)* |
| $R_{merge}$ | 0.11 (.98) |
| $R_{pim}$ | 0.07 (.72) |
| $I / \sigma I$ | 7.9 (1.6) |
| *CC (1/2)* | 0.99 (.63) |
| Completeness (%) | 100 (100) |
| Redundancy | 5.5 (5.5) |
| **Refinement** | |
| Resolution (Å) | 38.5–2.0 (2.05–2.00) |
| No. reflections / $R_{free}$ Reflections | 37,357 / 3724 |
| $R_{work} / R_{free}$ | 0.208/0.238 |
| No. atoms | |
| Protein | 3257 |
| Ligand/Solvent | 252 |
| *B*-factors | |
| Protein | 42 |
| Ligand/Solvent | 44 |
| Average | 42 |
| R.m.s. deviations | |
| Bond lengths (Å) | 0.007 |
| Bond angles (°) | 0.96 |

*Values in parentheses are for highest-resolution shell.

other CDRs show electron density for a sulfate group (*Figure 1—figure supplement 1*). By contrast, the CRH3s of other V1V2 bNAbs form stable subdomains involving regions of two-stranded β-sheet (*Figure 1E*), sometimes containing sulfated tyrosine sidechains (e.g., PG9's Tyr100G$_{CDRH3}$ and Tyr100H$_{CDRH3}$) that interact with basic residues within V1V2 (*McLellan et al., 2011*; *Pancera et al., 2013*; *Liu et al., 2017*).

The CDRH2 loop of BG1 is two residues longer than CDRH2s of other V1V2 bNAbs (*Figure 1A*), and it extends further from the antigen-binding site than the CDRH2 loops of other V1V2 bNAbs. (*Figure 1E*). The combined effects of the relatively compact CDRH3 and protruding CDRH2 result in the BG1 antigen-binding site being more flat than the combining sites of other V1V1 bNAbs (*Figure 1E*). The BG1 CDRH2 includes a solvent-exposed tryptophan at position 52B (*Figure 1D*) (lysine in the VH3-49*04 germline sequence from which BG1 is derived). Protein-protein interfaces often include surface-exposed hydrophobic residues (*Dall'Acqua et al., 1996*; *Kelley and O'Connell, 1993*; *Wells and de Vos, 1996*), suggesting that the long CDRH2, and this residue in particular, could be involved in the binding interface of BG1 with Env.

## Two BG1 Fabs bind per Env trimer

To characterize the BG1 interaction with Env, we determined the stoichiometry of binding of the BG1 Fab and Env trimer using size exclusion chromatography with multi-angle light scattering (SEC-MALS) and negative-stain EM single-particle analysis. SEC-MALS can be used to determine the stoichiometry of a complex by deriving the absolute molecular mass of the complex independent of shape and model (*Wyatt, 1993*). In the SEC-MALS experiment, we incubated a fully glycosylated soluble native-like Env trimer (BG505 SOSIP.664 [*Sanders et al., 2013*]; hereafter called Env) with a 3-fold molar excess of a Fab from a V1V2 bNAb or with a Fab from the G52K5 variant of 8ANC195, a gp120-gp41–spanning bNAb (*Scharf et al., 2014*) (calculated assuming a molecular mass of 210 kDa for Env trimer). The molecular mass of the PG16 Fab–Env complex was consistent with a stoichiometry of 1 Fab per Env trimer (*Figure 2A*), as previously shown by SEC-MALS and negative-stain EM for PG9 (*Julien et al., 2013b*), and the 8ANC195–Env complex stoichiometry was 3 Fabs per trimer (*Figure 2A*), consistent with crystal and EM structures of 8ANC195–BG505 Env complexes (*Scharf et al., 2015*; *Wang et al., 2016*). By contrast, the SEC-MALS profile for the BG1 Fab–Env complex indicated a stoichiometry of 2 Fabs per trimer (*Figure 2A*).

To confirm the conclusions from the SEC-MALS experiments, we examined the structures of BG1-Env-8ANC195, PG9-Env-8ANC195, and PG16-Env-8ANC195 complexes by negative-stain EM. Single-particle 3D reconstructions of these complexes at resolutions between 27 Å and 32 Å resolution (*Figure 2B*; *Figure 2—figure supplement 1*) revealed three 8ANC195 Fabs bound to BG505 Env trimer in the expected positions at the gp120-gp41 interface (*Scharf et al., 2015*; *Wang et al., 2016*; *Scharf et al., 2014*) of each complex. The BG1-Env-8ANC195 complex showed density for two BG1 Fabs per Env trimer, while the PG9-Env-8ANC195 and PG16-Env-8ANC195 complexes each showed density for one Fab per trimer (*Figure 2B*). The positions of the single PG9 and PG16 Fabs at the apex of Env trimer were similar to each other and to a previous negative-stain PG9-Env structure (*Julien et al., 2013b*), whereas the BG1 Fabs bound Env in an orientation distinct from the orientations of PGT145 (*Liu et al., 2017*; *Lee et al., 2017*), PG9, and PG16 (*Figure 2C*).

## Comparison of neutralization potencies of Fab and IgG forms of BG1

The close proximity of two BG1 Fabs at the apex of Env trimer (*Figure 2B*) raised the possibility that BG1 IgG binds bivalently to Env trimer, which would result in increased avidity due to intra-spike crosslinking (*Klein and Bjorkman, 2010*; *Galimidi et al., 2015*). To evaluate this possibility, we compared the neutralization potencies of IgG and Fab forms of BG1 across multiple viral strains using in vitro neutralization assays (*Montefiori, 2005*) and calculated the molar neutralization ratio (MNR) for BG1 (defined as [IC$_{50}$ Fab (nM)/IC$_{50}$ IgG (nM)]; [*Klein and Bjorkman, 2010*]). This ratio would be 2.0 in the absence of avidity effects from intra-spike crosslinking or inter-spike crosslinking (which we argued is rare due to the low numbers and densities of spikes on HIV-1 virions [*Klein and Bjorkman, 2010*; *Galimidi et al., 2015*]), and no steric effects that would increase neutralization potencies for the larger IgGs versus the smaller Fabs (*Klein and Bjorkman, 2010*). Viruses with densely-packed spikes can exhibit MNRs over 1000 (*Klein and Bjorkman, 2010*), whereas mean MNRs for anti-HIV-1 bNAbs tend to be low (*Klein and Bjorkman, 2010*; *Galimidi et al., 2015*; *West et al., 2012*).

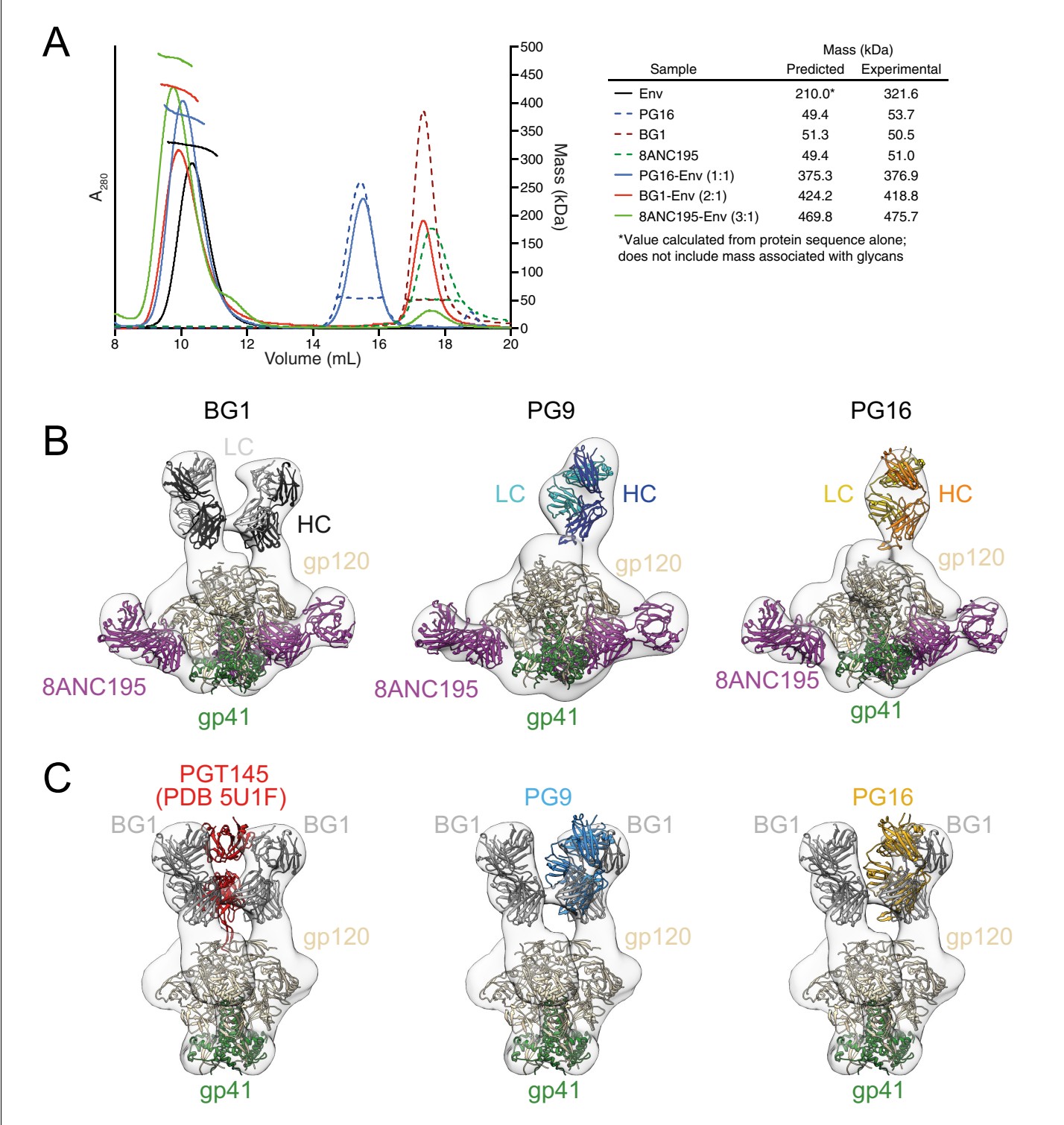

**Figure 2.** Fab-Env binding stoichiometries for V1V2 bNAbs. (**A**) SEC-MALS profiles for BG505 SOSIP.664 Env trimer and complexes of Env trimer with BG1, PG16, and 8ANC195 Fabs. The absorbance at 280 nm (left y-axis) is plotted against elution volume from a Superdex 200 10/300 GL gel filtration column and overlaid with the molar mass determined for each peak (right y-axis). Predicted and calculated molecular masses are shown in the table. (**B**) Negative-stain single particle EM reconstructions BG1-Env-8ANC195, PG9-Env-8ANC195, and PG16-Env-8ANC195 complexes. EM density was fit with coordinates for the indicated Fabs and for BG505 Env trimer. (**C**) Density and coordinates from the BG1-Env portion of the BG1-Env-8ANC195 reconstruction (density for 8ANC195 Fabs removed) with coordinates for the indicated Fabs superimposed. The Env trimer portion from EM structures

*Figure 2 continued on next page*

*Figure 2 continued*
of complexes of PGT145-Env (PDB 5U1F), PG9-Env (panel B), or PG16-Env (panel B) were superimposed with the Env trimer portion of the BG1-Env-8ANC195 structure (panel B).
The following source data and figure supplement are available for figure 2:

**Source data 1.** Molar neutralization ratios (MNRs) for select bNAbs (MNR = [IC$_{50}$ Fab (nM)/IC$_{50}$ IgG (nM)].
**Figure supplement 1.** Negative-stain EM structures of V1V2 bNAb-Env-8ANC195 complexes.

We found that the geometric mean MNR for BG1 was 8.4, sd$_{geom}$ (geometric standard deviation) = 2.4 (*Figure 2—source data 1*). To put this result into context, we derived MNRs using published data (*Galimidi et al., 2015*; *West et al., 2012*) for bNAbs that should not be capable of bivalent binding to a single Env trimer: PG9 and PG16 (because only one Fab binds per trimer), and the CD4bs bNAbs VRC01 and 3BNC60, whose adjacent epitopes on neighboring gp120s are located in positions that would not permit simultaneous binding by both Fabs of an IgG. The mean MNRs for these bNAbs were comparable to the mean MNR for BG1 (PG9: 6.0, sd$_{geom}$ = 4.1; PG16: 10.3, sd$_{geom}$ = 3.4; VRC01: 4.8, sd$_{geom}$ = 3.1; 3BNC60: 8.1, sd$_{geom}$ = 3.2) (Figure 2—figure supplement 2). By contrast, the mean MNR for a 3BNC60-based reagent that exhibited neutralization potency increases suggestive of intra-spike crosslinking (a bivalent form of 3BNC60 in which two 3BNC60 Fabs were joined by a double-stranded DNA linker [*Galimidi et al., 2015*]) was 141, sd$_{geom}$ = 2.1. These results suggest that, like other natural HIV-1 bNAbs, BG1 IgG does not bind bivalently to Env trimers during neutralization.

## Cryo-EM structures of BG1-Env-8ANC195 and PG9-Env-8ANC195 complexes

In order to explore the details of how two BG1 Fabs bind asymmetrically to the three V1V2 loops in an Env trimer, we used single-particle cryo-EM to solve the structure of a BG1-Env complex. A single-particle reconstruction of a BG1-Env-8ANC195 complex with two BG1 Fabs per Env trimer was obtained at a resolution of ~6.2 Å (calculated using the 0.143 gold-standard Fourier shell coefficient cutoff criterion) (*Scheres and Chen, 2012*) (*Figure 3*; *Figure 3—figure supplements 1* and *2*; *Video 1*; *Table 2*). To interpret the structure, the coordinates of BG1 Fab (this study) and 8ANC195 Fab (PDB code 4P9M) were fit by rigid body docking into the cryo-EM densities, after which the coordinates for three gp41 monomers (PDB 5FUU) (*Lee et al., 2016*) and three gp120 monomers (PDB 5T3X) (*Gristick et al., 2016*) were fit independently (*Figure 3A*). Coordinates for ordered N-linked glycans from a natively-glycosylated BG505 Env trimer structure (PDB 5T3X) (*Gristick et al., 2016*) were fit separately as rigid bodies at potential N-linked glycosylation sites (PNGSs) at which EM density was apparent (*Figure 3B*). After initial rigid body docking, refinement of the protein portions of the complex was carried out with geometric restraints for protein and N-glycan residues (*Adams et al., 2010*; *Agirre et al., 2015*). Unlike the PG9 Fab, in which the CDRH3 loop is ordered only when bound to the V1V2 scaffold (*McLellan et al., 2011*), the CDRH3 of BG1 was ordered in both the unbound (*Figure 1E*) and bound (*Figure 3C*) states. Within the limits of the resolution of the cryo-EM structure, the CDRH3 and CDRH2 loops of BG1_A and BG1_B adopted the same conformation as their counterparts in the unbound BG1 crystal structure (*Figure 3C,D*). In particular, the relatively compact CDRH3 loop observed in the unbound BG1 crystal structure (*Figure 1E*) was maintained in the 2:1 BG1-Env structure (*Figure 3C*).

A 27 Å BG1-Env-8ANC195 complex with three BG1 Fabs per Env trimer (3:1 BG1-Env) was also reconstructed from a minor subset of the total particles (<10%) (*Figure 3—figure supplement 1*), and coordinates were fit to the density by rigid body refinement (*Figure 3—figure supplement 2*).

We also solved an 11.5 Å single-particle cryo-EM reconstruction of a PG9-Env-8ANC195 complex (*Figure 4*; *Figure 4—figure supplement 1*). As described for the BG1-Env-8ANC195 cryo-EM structure, coordinates of PG9 Fab (PDB 3U36) were fit, after which gp120 and gp41 subunits were fit independently. Since the CDRH3 of unliganded PG9 Fab (PDB 3U36) is disordered, we modeled the unliganded PG9 coordinates and the ordered CDRH3 from the PG9 Fab in the PG9-V1V2 scaffold structure (PDB 3U4E) (*McLellan et al., 2011*) separately into the EM density. The position of the

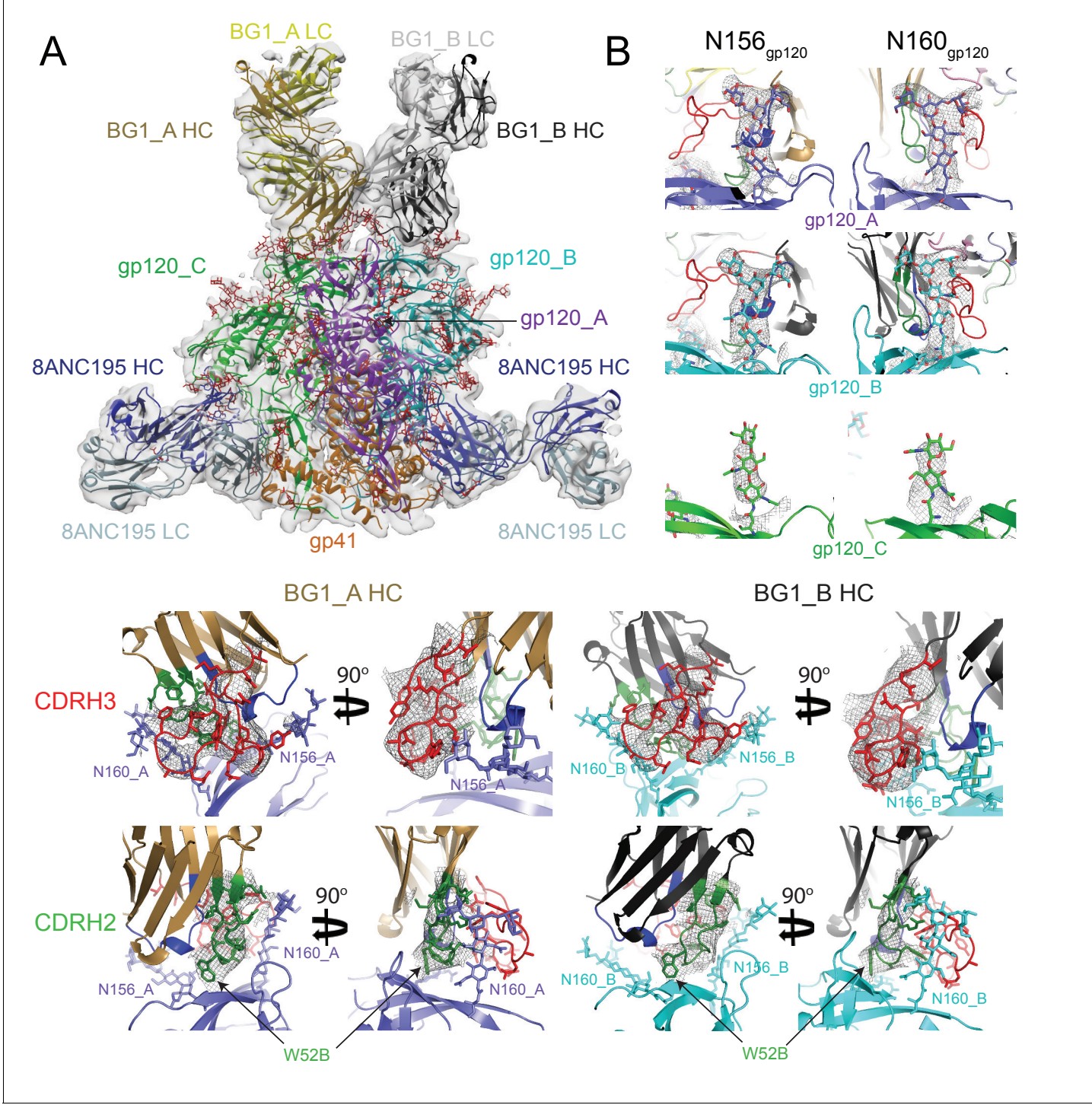

**Figure 3.** ~6.2 Å cryo-EM reconstruction of a BG1-Env-8ANC195 complex. (**A**) EM density fit by coordinates for BG1_A (HC in brown; LC in yellow), BG1_B (HC in dark gray; LC in light gray), gp120s associated with BG1_A and BG1_B Fabs in purple and cyan, respectively; gp120_C in green, gp41 in orange, and 8ANC195 Fabs in dark blue (HC) and light blue (LC). (**B**) Close-up views of densities contoured at 6.0σ (0.0378 e/Å$^3$) for glycans attached to Asn156$_{gp120}$ and Asn160$_{gp120}$ in the three gp120 protomers of Env trimer. Glycan residues built into the cryo-EM map densities are shown as sticks with oxygen atoms in red and nitrogen atoms in blue. (**C, D**) Close-up views of densities contoured at 6.0σ (0.0378 e/Å$^3$) surrounding CDRH3 (panel C) and CDRH2 (panel D) in BG1-gp120_A and BG1_B-gp120_B interactions. Asn156$_{gp120}$ and Asn160$_{gp120}$ glycans are shown as sticks. CDRH2 residue Trp54 (*Figure 1D*) is identified with an arrow.

The following figure supplements are available for figure 3:

*Figure 3 continued on next page*

*Figure 3 continued*

**Figure supplement 1.** Cryo-EM data processing for the BG1-Env-8ANC195 complex structure.

**Figure supplement 2.** Validation of the 2:1 and 3:1 BG1-Env structures.

single PG9 Fab on Env trimer in the cryo-EM structure was consistent with negative-stain structures reported here and previously (*Julien et al., 2013b*). The angle of approach of the PG9 Fab was different in the PG9-Env interaction predicted from modeling the PG9 Fab-monomeric V1V2 scaffold structure (*McLellan et al., 2011*) onto Env trimer versus in the PG9-Env EM structure (*Figure 4B*), likely due to accommodating glycans in a neighboring gp120 protomer in the trimeric Env that are not present in a monomeric V1V2 scaffold.

## Comparison of BG1 and PG9 epitopes and paratopes

The 6.2 Å BG1-Env-8ANC195 structure with two BG1 Fabs (2:1 BG1-Env) showed interactions with two of the three gp120 protomers within Env trimer. To distinguish the structurally-distinct gp120 and BG1 subunits within the 2:1 BG1-Env structure, we defined the two gp120 subunits of Env that interact with BG1 Fabs as gp120_A and gp120_B, the corresponding BG1 Fabs as BG1_A and BG1_B, and the third gp120 subunit, which showed no contacts with BG1 Fabs, as gp120_C (*Figure 3A*; *Figure 5A*).

To compare the BG1-Env and PG9-Env interactions, we calculated approximate footprints of the each antibody on Env trimer (epitopes) and of Env on each antibody (paratopes). For the BG1-Env interaction, we used coordinates fit to the 6.2 Å BG1-Env structure, and for the PG9-Env interaction, we used coordinates fit to the 11.5 Å PG9-Env structure. To account for low resolution and/or modeling errors, we assigned contacts using a criterion of a distance of $\leq$7 Å between antibody-antigen $C\alpha$ atoms for protein-protein contacts and $\leq$7 Å between protein $C\alpha$ and glycan C1 atoms for antibody-antigen-glycan contacts. As shown in *Figure 5A*, the contact analysis predicted that both Fabs reached through the glycan shield between the $Asn156_{gp120}$ and $Asn160_{gp120}$ glycans to contact underlying V1V2 protein residues, but only PG9 made contacts with basic residues in a lysine-rich region of the V2 loop ($Lys168_{gp120}$ and $Lys169_{gp120}$). This prediction was validated by an analysis of in vitro neutralization data demonstrating that substitution of $Lys169_{gp120}$ reduces neutralization potency for PG9, but has only minor effects on BG1 (*Figure 5—figure supplement 1A*). Conversely, substitution of $Thr132_{gp120}$ had more of an effect on BG1 neutralization than on PG9 neutralization (*Figure 5—figure supplement 1B*).

Predicted protein-protein contacts were mediated by different CDR loops for BG1 versus PG9: CDRH2 for BG1 and CDRH3 for PG9. Contacts with Env glycans were also made differently for BG1 and PG9: The $Asn156_{gp120}$ glycan on both gp120_A and gp120_B was contacted by the BG1 CDRL1, and the $Asn160_{gp120}$ glycan was contacted on both gp120s by the CDRH2 and CDRH3 loops of BG1 and also by the CDRL3 in the case of BG1_B. By contrast, PG9 contacted only the $Asn160_{gp120}$ glycan on gp120_B, and these contacts were mediated exclusively by CDRH3. Examination of the paratopes on BG1 versus PG9 illustrated that BG1 uses several CDRs for interacting with Env, whereas PG9 uses only its protruding CDRH3 (*Figure 5B*; *Figure 6A,B*). In particular, Trp52B within CDRH2 of the BG1 heavy chain is in the

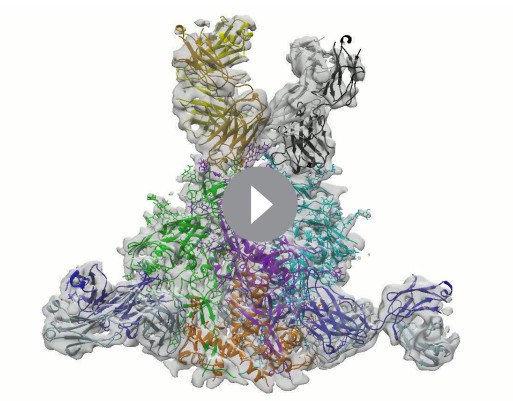

**Video 1.** ~6.2 Å Cryo-EM reconstruction of BG1-Env-8ANC195 complex. Color scheme follows *Figure 3A*: BG1 Fabs are shades of yellow (BG1_A) and gray (BG1_B); gp120s are purple (gp120_A), cyan (gp120_B), and green (gp120_C); gp41 is orange, and 8ANC195 Fabs are shades of blue.

**Table 2.** BG1-Env-8ANC195 complex cryo-EM structure data collection and model statistics.

| Data collection/processing | |
| --- | --- |
| Microscope | Titan krios |
| Voltage | 300 kV |
| Camera | Gatan K2 Summit |
| Camera mode | counting |
| Defocus range | 1.5–3.5 |
| Exposure time | 15 s |
| Dosage rate | 2.92 electrons·pixel$^{-1}$·subframe$^{-1}$ |
| Magnified pixel size | 1.35 |
| Total Dose (e/A2) | 80 |
| Reconstruction | |
| Software | Relion-1.4 |
| Symmetry | C1 |
| Particles refined | |
| Resolution(automask, Å) | 6.18 |
| Resolution(masked out Fab CHCL domains, Å) | 6.18 |
| Map sharpening B-factor (Å2) | −24.7274 |
| Model Statistics | |
| Map CC (whole unit cell): | 0.931 |
| Map CC (around atoms): | 0.731 |
| All-atom clashscore | 17.66 |
| Ramachandran plot: | |
| outliers: | 0.44% |
| allowed: | 7.44% |
| favored: | 92.12% |
| rmsd (bonds): | 0.01 |
| rmsd (angles): | 1.11 |
| Rotamer outliers: | 0.85% |
| C-beta deviations: | 0 |

vicinity of the protein portion of the V1V2 epitope (*Figure 3D*).

Comparison of the BG1 and PG9 epitopes on Env trimer rationalize differences in their relative neutralization potencies across HIV-1 strains. To find Env sequence features that contribute to BG1 potency variations, we analyzed in vitro neutralization results using previously-described methods (*West et al., 2013*). The analysis revealed that the presence of a potential N-linked glycosylation site (PNGS) at position 130 of gp120 correlated with weaker BG1 neutralization (geometric mean $IC_{50}$ = 25.7 µg/mL for strains including PNGS 130, versus 1.5 µg/mL for strains lacking PNGS 130) (*Figure 5A*; *Figure 5—source data 1*). PG9 was also more potent on strains lacking PNGS 130 (geometric mean $IC_{50}$ = 0.76 µg/mL for strains including PNGS 130, versus 0.24 µg/mL for strains lacking PNGS 130); however, the difference was larger for BG1 (17-fold for BG1 versus 3-fold for PG9). This result is rationalized by the BG1 and PG9 epitopes on Env trimer: although residue 130 ($Gln130_{gp120}$) is not part of a PNGS in the BG505 strain of Env in our structural studies, residue 130 lies directly within the protein region contacted by BG1, but is outside of the PG9 footprint (*Figure 5A*), suggesting that addition of an N-glycan at position 130 would more directly disrupt BG1 binding than PG9 binding.

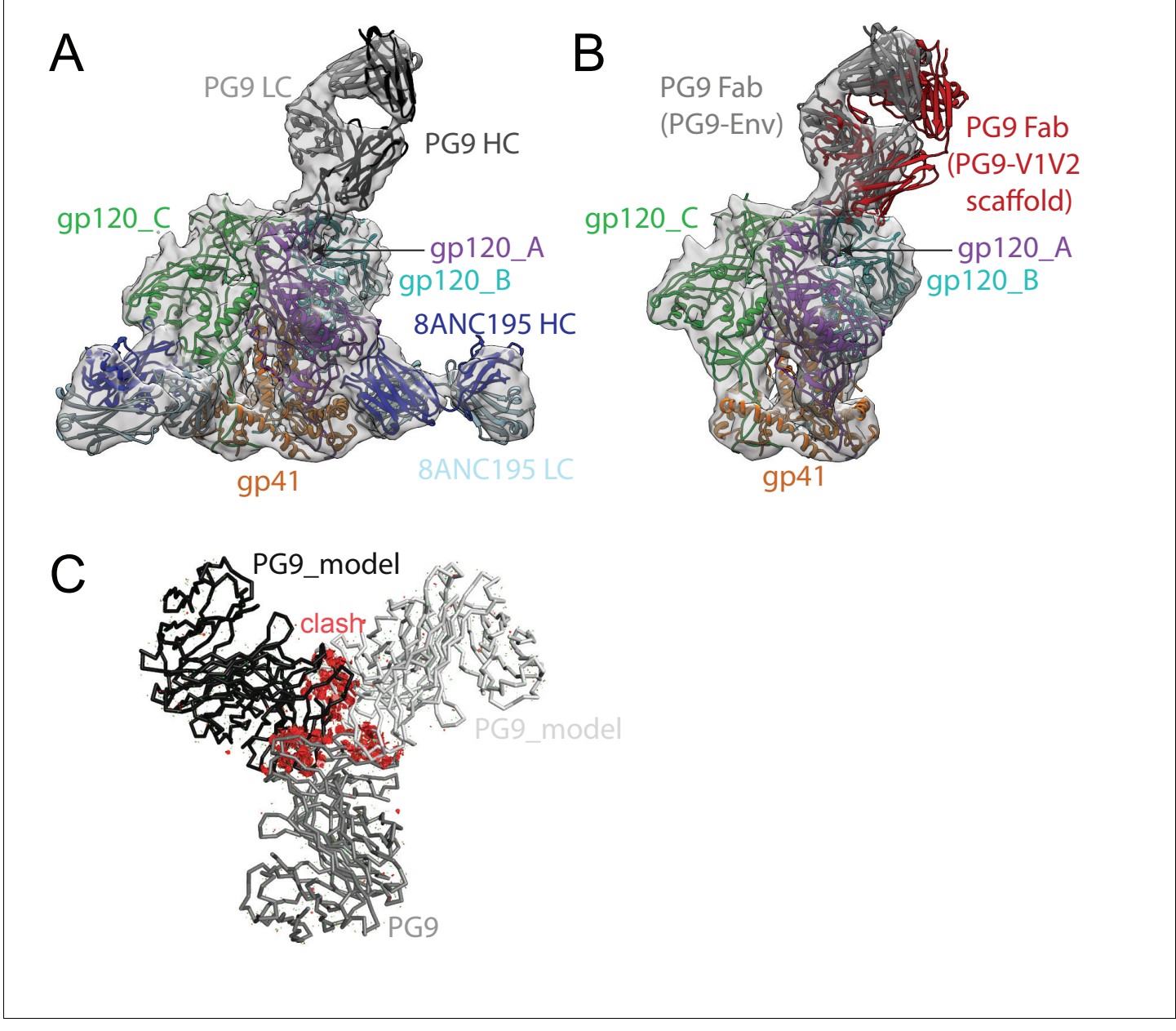

**Figure 4.** ~11.5 Å cryo-EM reconstruction of a PG9-Env-8ANC195 complex. (**A**) EM density fit by coordinates for PG9 (HC in dark gray; LC in light gray), gp120 subunits in dark blue, cyan, and green, gp41 in orange, and 8ANC195 Fabs in dark blue (HC) and light blue (LC). (**B**) Density and coordinates from the PG9-Env portion of the PG9-Env-8ANC195 reconstruction (density for 8ANC195 Fabs removed) with coordinates for the PG9 Fab from the PG9-V1V2 scaffold structure (PDB 3U4E) superimposed. When the V1V2 scaffold from the PG9-V1V2 complex structure was superimposed with the V1V2 portion of the PG9-Env-8ANC195 structure (panel A), the angle of approach of the PG9 Fab differs from the angle observed in the PG9-Env structure. (**C**) Hypothetical model of three PG9 Fabs bound per Env trimer. One PG9 Fab (PG9_A) is shown its experimentally-determined position from the PG9-Env-8ANC195 structure interacting mainly with gp120_A on Env trimer (view looking down the trimer three-fold axis). The $V_H$-$V_L$ domains for second and third PG9 Fabs (PG9 model) were positioned onto the gp120_B and gp120_C subunits assuming the interaction observed for PG9_A with gp120_A. Predicted van der Waals clashes (red dots) were calculated using the show_bumps module in Pymol (*Schrödinger, 2011*).

The following figure supplement is available for figure 4:

**Figure supplement 1.** Cryo-EM data processing for PG9-Env-8ANC195 complex structure.

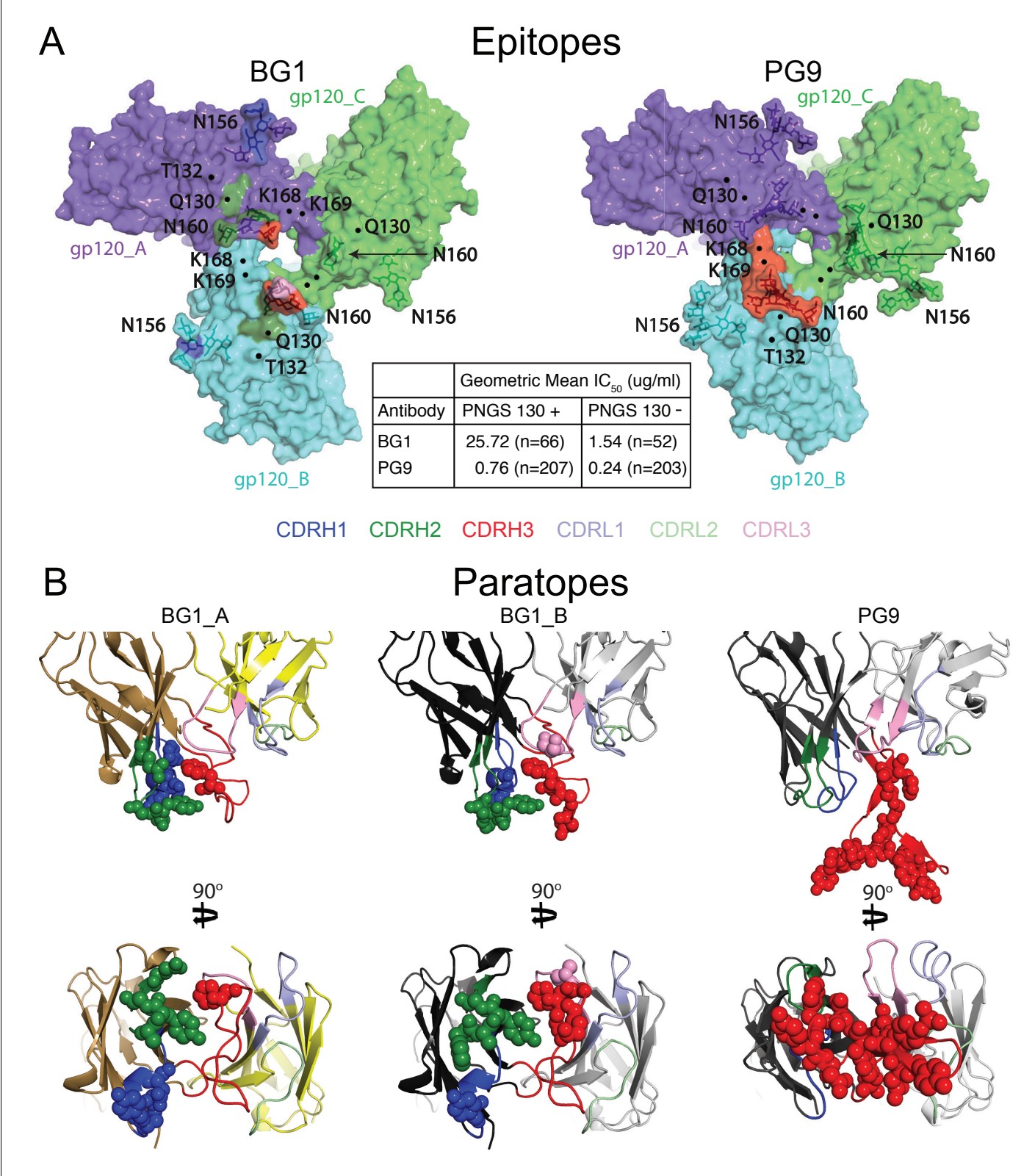

**Figure 5.** Comparison of BG1 and PG9 epitopes and paratopes. (**A**) Epitopes on Env trimer (top view surface representations) were defined as protein or glycan residues whose Cα (for protein) or C1 (for glycans) atom was within 7 Å of the bound Fab. Contacts on Env are color-coded to indicate which CDR loop made the contact (CDRH3 in red, CDRH2 in green, and CDRH1 in dark blue). Glycans are shown as sticks with a nearby label to identify the glycan as attached to either Asn156$_{gp120}$ or Asn160$_{gp120}$, and locations of residues of interest (Lys168$_{gp120}$, Lys169$_{gp120}$, and Gln130$_{gp120}$) are indicated

*Figure 5 continued on next page*

Wang *et al.* eLife 2017;6:e27389. DOI: 10.7554/eLife.27389

*Figure 5 continued*

by black dots. Center box compares geometric mean $IC_{50}$ values for BG1 and PG9 IgGs evaluated against HIV-1 strains either containing or not containing a PNGS at position 130 (number of HIV-1 strains indicated in the parentheses). $IC_{50}$ values > 50 µg/mL set to 50 µg/mL for geometric mean calculations. (B) Paratopes on BG1_A, BG1_B, and PG9 Fabs indicated by spheres on ribbon representations of $V_H$-$V_L$ domains shown in two orientations related by a 90° rotation. CDRs are color coded as in panel A.

The following source data and figure supplement are available for figure 5:

**Source data 1.** In vitro neutralization data for BG1 and PG9 strains with and without N130 glycan.

**Figure supplement 1.** Coverage curves for results of in vitro neutralization across panels of viruses that do or do not include a lysine at position $169_{gp120}$ (panel A) or a threonine at position $132_{gp120}$ (panel B).

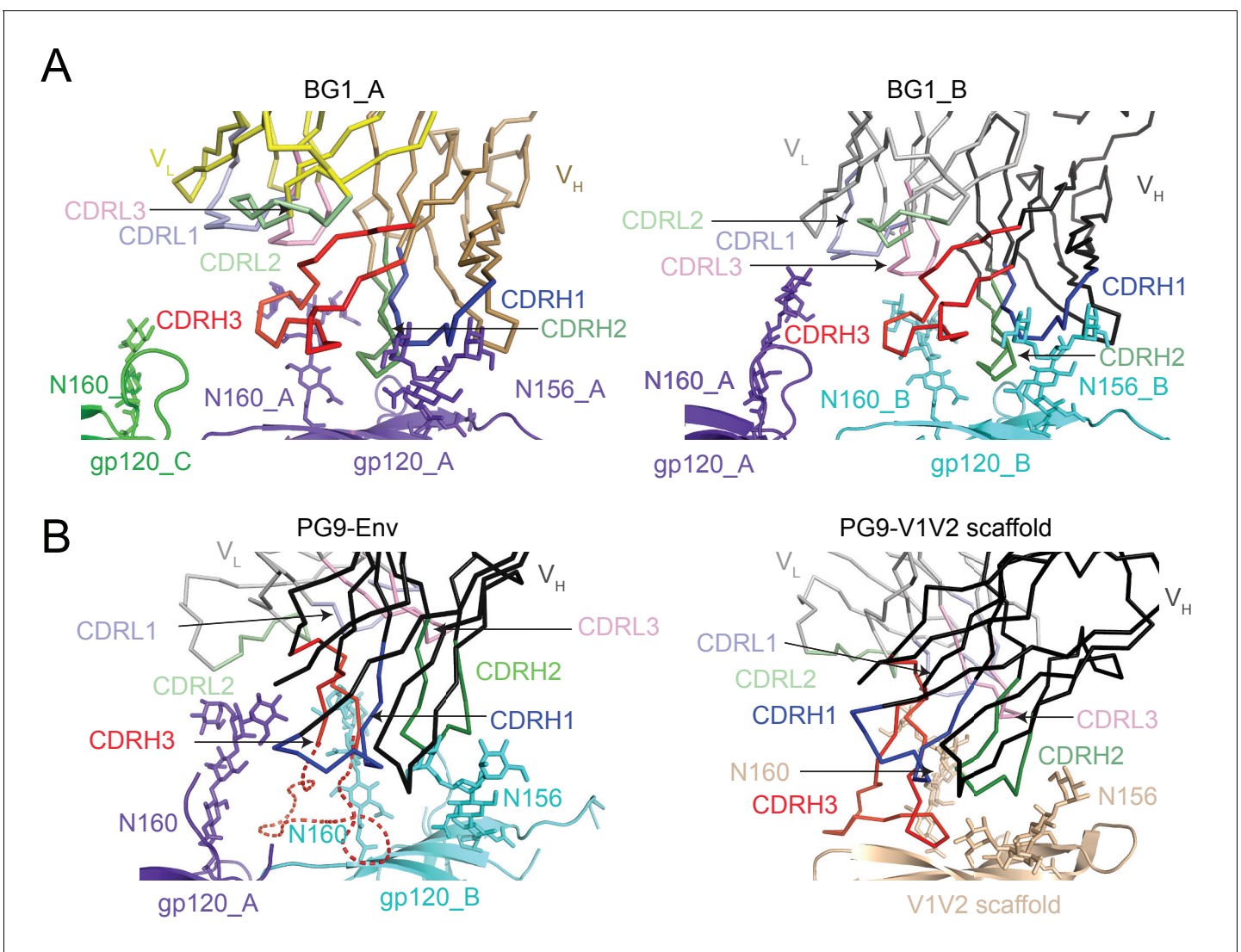

**Figure 6.** BG1 and PG9 interactions with V1V2. (A) BG1 interactions with N-linked glycans attached to $Asn156_{gp120}$ and $Asn160_{gp120}$ shown for BG1_A (left) and BG1_B (right). (B) PG9 interactions with N-linked glycans attached to $Asn156_{gp120}$ and $Asn160_{gp120}$ from PG9-Env cryo-EM structure reported here (left) and from the crystal structure of PG9-V1V2 scaffold (PDB 3U4E) (right). The position of the CDRH3 loop in the PG9-Env cryo-EM structure is shown by a dashed line representing the position of CDRH3 in the PG9-V1V2 scaffold structure.

## Asymmetry of BG1-Env and PG9-Env complexes

To identify important common features of the BG1_A–gp120_A and BG1_B–gp120_B interactions, we aligned the two BG1-gp120 complexes by superimposing either the protein coordinates of the two V1V2 regions (*Figure 7A*) or by superimposing the $Asn156_{gp120}$ and $Asn160_{gp120}$ glycans of gp120_A and gp120_B (*Figure 7B*). When aligning the protein coordinates of the two V1V2 regions, the BG1 $V_H$-$V_L$ domains superimposed with a root mean square deviation (rmsd) of 7.7 Å for 435 Cα atoms. The rmsd for the same number of Cαatoms decreased to 2.9 Å when we superimposed the $Asn156_{gp120}$ and $Asn160_{gp120}$ glycans of the two V1V2 regions. This comparison suggests that glycan contacts play a larger role than protein contacts in determining how BG1 recognizes the V1V2 epitope on Env.

To address why the 6.2 Å BG1-Env-8ANC195 structure shows two BG1 Fabs arranged asymmetrically on Env trimer, rather than three symmetrically-arranged BG1 Fabs, we used modeling to construct an Env complex containing a third BG1 Fab. The $V_H$-$V_L$ domains for a third BG1 Fab (BG1 model) were positioned onto the gp120_C subunit assuming that a third BG1 Fab would interact with the $Asn156_{gp120}$ and $Asn160_{gp120}$ glycans in the common interaction observed for BG1_A and BG1_B (*Figure 7B*). Predicted van der Waals clashes (red dots) demonstrated that a third BG1 Fab cannot be accommodated on the Env trimer structure without steric clashing (*Figure 7C*). However, as indicated by the minor population of Env trimers with three BG1 Fabs in the 3:1 BG1-Env structure (*Figure 3—figure supplement 2*), binding of three Fabs is possible (*Figure 7D*). When three Fabs were bound, the Env trimer appeared more open than Env in the 2:1 BG1-Env structure, as shown by a comparison of the BG1 Fab positions in the 2:1 and 3:1 BG1-Env structures (*Figure 7E*).

## Discussion

The V1V2 epitope on HIV Env is targeted by bNAbs that make quaternary interactions that prevent opening of Env trimer to expose the V3 loop and coreceptor binding site, hence blocking conformational changes leading to fusion of the viral and host cell membranes (*McLellan et al., 2011*; *Pancera et al., 2013*; *Pan et al., 2015*; *Walker et al., 2011*, *2009*; *Bonsignori et al., 2011*; *Doria-Rose et al., 2014*). Quaternary interactions visualized thus far for V1V2 bNAbs involve the binding of a single Fab to the apex of Env trimer (*Julien et al., 2013b*; *Liu et al., 2017*), but here we demonstrate that the stoichiometry of binding for the new V1V2 bNAb BG1 (*Freund et al., 2017*) is two Fabs per Env trimer, with a minor population of 3:1 BG1-Env complexes (*Figures 2* and *3*; *Figure 3—figure supplement 1*).

We previously noted that epitopes for HIV-1 bNAbs are located in positions not predicted to allow simultaneous binding of both Fabs from an IgG to a single Env trimer (*Klein and Bjorkman, 2010*; *Galimidi et al., 2015*). By definition, the two Fabs in IgG forms of V1V2 bNAbs that exhibit a 1 Fab per Env trimer stoichiometry cannot bind with avidity to an Env trimer, and epitopes on HIV-1 Env that show a 3:1 Fab-Env stoichiometry are located in positions that are too far apart to allow simultaneous binding of both Fabs in an IgG to one Env (*Klein and Bjorkman, 2010*; *Galimidi et al., 2015*). However, the V1V2 bNAb BG1 binds with a 2:1 Fab-Env stoichiometry to an epitope in close proximity on neighboring gp120 subunits at the apex of Env, suggesting that it might be possible for both Fabs of a single BG1 IgG to bind simultaneously to a single Env trimer. However, comparisons of neutralization potencies of BG1 Fab and IgG suggested that, as found for other HIV-1 bNAbs, potential avidity effects for BG1 IgG were marginal (Figure 2—figure supplement 2), consistent with no intra-spike crosslinking and only minimal inter-spike crosslinking or steric effects favoring an IgG over a Fab. The apparent absence of intra-spike crosslinking for BG1 IgG can be rationalized by the angle of approach the Fabs adopted in the 2:1 BG1-Env structure: adjacent Fabs were oriented such that they point away from each so that the C-termini of the $C_H1$ domains (to which the N-terminus of an IgG Fc would be attached) are too far apart (~85 Å) to be spanned by a natural IgG hinge region (e.g., the C-termini of the two Fabs in the crystal structure of intact IgG b12 (*Saphire et al., 2001*) (PDB 1HZH) are separated by ~18 Å). Thus it appears that HIV-1 Env has evolved to prevent bivalent binding by IgGs to Env trimer even when Fabs from two separate IgGs can bind to nearby epitopes.

Previously-characterized V1V2 bNAbs made primary interactions with Env that involved a long, negatively-charged CDRH3 that reached through the Env glycan shield to contact basic residues in the V2 loop (*McLellan et al., 2011*; *Pancera et al., 2013*; *Pan et al., 2015*; *Julien et al., 2013b*;

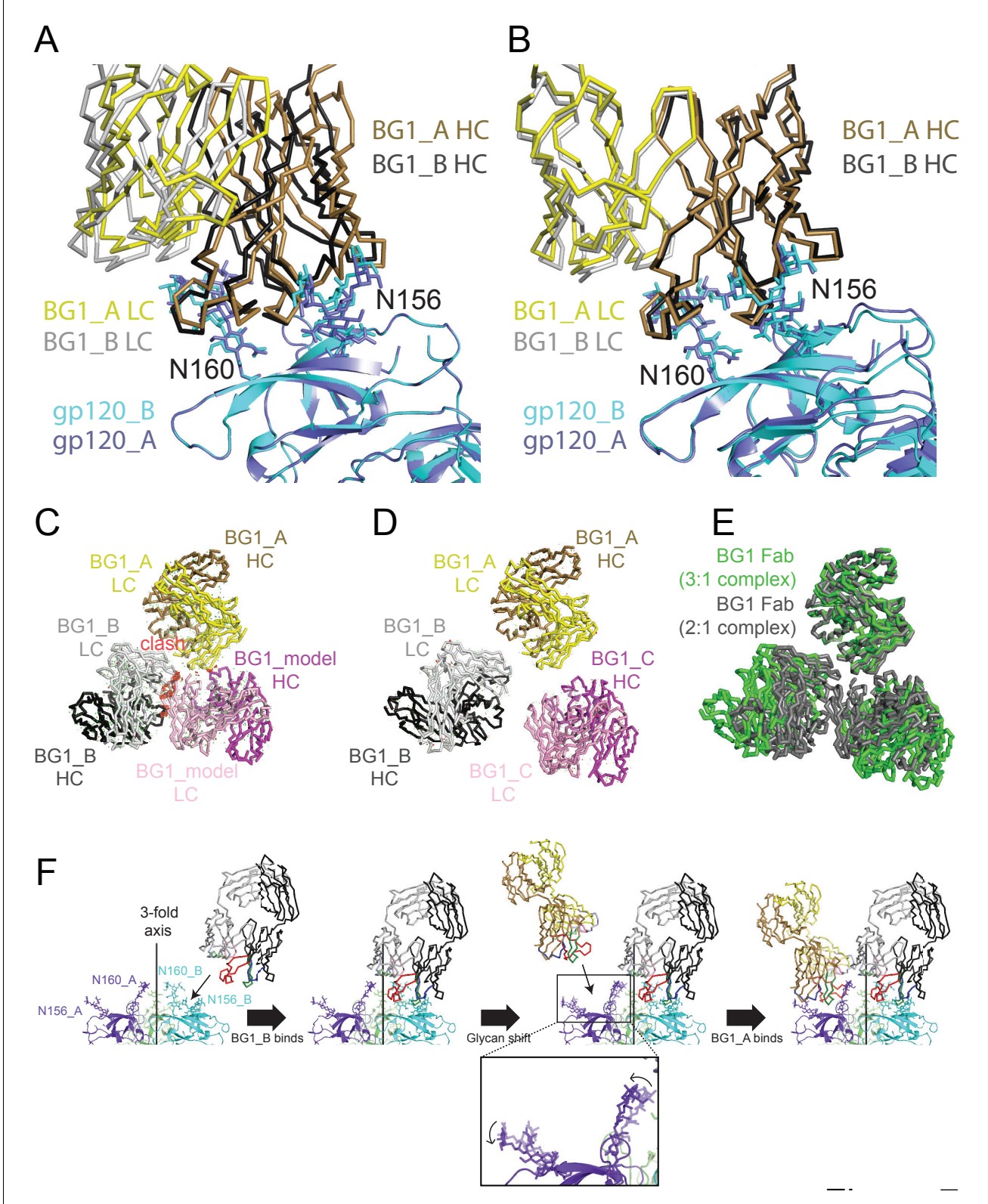

**Figure 7.** Asymmetry of BG1 interactions with Env trimer. (**A,B**) Superposition of the two BG1 binding sites in the 2:1 BG1-Env structure by aligning either the V1V2 regions of gp120_A and gp120_B (panel A) or the Asn156$_{gp120}$ and Asn160$_{gp120}$ glycans within gp120_A and gp120_B (panel B). (**C**) Hypothetical model of three BG1 Fabs bound per Env trimer based on Fab orientations in the 2:1 BG1-Env structure. The BG1_A and BG_1 B Fabs are shown in their respective positions on the gp120_A and gp120_B subunits of Env trimer (view looking down the trimer three-fold axis). A third BG1 Fab

*Figure 7 continued on next page*

*Figure 7 continued*

(BG1 model) was positioned onto the gp120_C subunit assuming that it would interact with the $Asn156_{gp120}$ and $Asn160_{gp120}$ glycans in the common interaction observed for BG1_A and BG1_B (panel B). Predicted van der Waals clashes (red dots) were calculated using the show_bumps module in Pymol (*Schrödinger, 2011*). (D) Positions of three BG1 Fabs in the 27 Å 3:1 BG1-Env structure (no predicted van der Waals clashes). (E) Comparison of BG1 Fab positions in the 2:1 BG1-Env structure (gray; the third Fab is from the model shown in panel C) and the 3:1 BG1-Env structure (green) after superimposing the BG1_A Fabs. (F) Model for binding of two BG1 Fabs to form the 2:1 BG1-Env structure. Panels 1 and 2: BG1_B binds first to the V1V2 epitope on gp120_B such that its CDRL1 interacts with the $Asn160_{gp120}$ glycan from gp120_A. Panels 3 and 4: BG1_A recognizes the $Asn156_{gp120}$ and $Asn160_{gp120}$ glycans from gp120_A. The flexible glycans are shifted away from the trimer 3-fold axis (indicated by arrows in the close-up view of the third panel) such that a potential clash with BG1_B is avoided.

*Liu et al., 2017*; *Walker et al., 2011*, *2009*; *Bonsignori et al., 2011*; *Doria-Rose et al., 2014*). Although BG1 is directed against the same general region of Env, its epitope shows glycan interactions dominating over protein interactions, and the relatively long BG1 CDRH3 (*Figure 1A*) is folded into a compact structure (*Figure 1D,E*) that is used to make contacts with Env glycans rather than Env protein residues (*Figure 5A*). The role of reaching through the glycan shield to contact protein residues is taken by BG1's unusually long CDRH2 (*Figure 1E*), resulting in a different epitope footprint on Env than seen for typical V1V2 bNAbs such as PG9 (*Figure 5A*).

Examination of the interactions of the two BG1 Fabs in the 2:1 BG1-Env complex solved by single particle cryo-EM suggests differences in the BG1_A and BG1_B interfaces with Env. The BG1_B Fab interacts not only with glycans on its primary subunit contact gp120_B; the complex structure also suggests a potential secondary interaction with the $Asn160_{gp120}$ glycan on a neighboring subunit, gp120_A, using CDRL1 (*Figure 6A*, right panel). By contrast to the BG1_B interactions with Env trimer, the 2:1 BG1-Env structure does not predict secondary contacts for BG1_A, a difference that may explain why BG1_A and BG1_B showed distinct angles of approach when we aligned the BG1_A–gp120_A and BG1_B–gp120_B complexes by superimposing the protein coordinates of the V1V2 regions (*Figure 7A*). Differences in the interactions of BG1_A and BG1_B with Env trimer suggests a model for BG1 Fab binding to Env (*Figure 7F*): If BG1_B binds first to the V1V2 epitope on gp120_B, its CDRL1 could then interact with the $Asn160_{gp120}$ glycan from gp120_A. After that, a second BG1 Fab, BG1_A, could recognize the $Asn156_{gp120}$ and $Asn160_{gp120}$ glycans from gp120_A. Because of the interaction between the BG1_B CDRL1 and the $Asn160_{gp120}$ glycan from gp120_A, the angle of approach of BG1_B is shifted further towards the Env trimer 3-fold axis as compared with the angle of approach of with BG1_A (*Figure 7C*; fourth panel). In this model, the intrinsic flexibility of N-linked glycans would allow BG1_A to capture an Env conformation in which the glycans are shifted away from the trimer 3-fold axis such that a potential clash with BG1_B is avoided. In this binding mode, a third BG1 Fab cannot bind because it would clash with BG1_B. However, if the Fabs can capture a more open state of Env trimer in the absence of CD4 binding, three Fabs can be accommodated (*Figure 7D,E*). The low percentage of 3:1 BG1-Env complexes observed by cryo-EM (*Figure 3—figure supplement 1*), despite incubation with a 9-fold molar excess of BG1 Fab (Materials and methods), suggests that formation of 3:1 BG1-Env complexes is rare.

Our direct comparisons of BG1-Env and PG9-Env stoichiometry and cryo-EM structures illustrate the diversity of bNAb recognition of the V1V2 epitope at the apex of HIV-1 Env trimer. These results provide new information relevant to design of immunogens to raise antibodies against V1V2 (*Escolano et al., 2017*), an important target because the V1V2 loops are critical for maintaining the closed Env trimer structure in which the coreceptor binding site is shielded and conformational changes leading to fusion of the viral and host cell membranes are prevented. The more typical form of V1V2 recognition, in which a protruding CDRH3 from, e.g., PG9, is used to reach through the $Asn156_{gp120}$ and $Asn160_{gp120}$ glycans to contact protein residues on the V2 loop, results in a more potent bNAb (*Figure 5A*; Figure 5—figure supplement 2). By contrast, the BG1 mode of V1V2 recognition, which is less potent, relies more on glycan contacts using relatively compact CDRs, and BG1's contacts with V1V2 protein residues are more easily disrupted by glycosylation at PNGS 130 (*Figure 5A*; Figure 5—figure supplement 2). Thus immunogens designed to raise bNAbs against the V1V2 epitope would likely require removal of the $Asn130_{gp120}$ glycan in order to target both BG1- and PG9-style V1V2 bNAbs.

## Material and methods

### Protein expression and purification

Fabs (BG1, PG9, PG16, and the G52K5 variant of 8ANC195 [*Scharf et al., 2014*]) and BG1 IgG were expressed and purified as described (*Scharf et al., 2016*). Briefly, IgGs and 6xHis-tagged Fabs were expressed by transient transfection of HC and LC expression plasmids into HEK293-6E cells. Fabs and BG1 IgG were purified from cell supernatants using Ni-NTA (GE Healthcare) (for Fabs) or protein A (GE Healthcare) (for IgG) affinity chromatography followed by SEC on a Superdex 200 16/60 column (GE Healthcare). Proteins were stored in 20 mM Tris, pH 8.0, and 150 mM sodium chloride (TBS buffer) supplemented with 0.02% (wt/vol) sodium azide.

Structural studies of Fab-Env complexes were done using BG505 SOSIP.664, a soluble clade A gp140 trimer that includes 'SOS' substitutions (A501C$_{gp120}$, T605C$_{gp41}$), the 'IP' substitution (I559P$_{gp41}$), the N-linked glycan sequence at residue 332$_{gp120}$ (T332N$_{gp120}$), an enhanced gp120-gp41 cleavage site (REKR to RRRRRR), and a stop codon after residue 664$_{gp4}$ (Env numbering according to HX nomenclature) (*Sanders et al., 2013*). Protein was expressed in HEK293-6E cells (National Research Council of Canada) in the absence of kifunensine by transient transfection of plasmids encoding BG505 SOSIP and soluble furin at a ratio of 4:1 as previously described (*Gristick et al., 2016*). We isolated BG505 SOSIP protein from cell supernatants using a 2G12 immunoaffinity column made by covalently coupling 2G12 IgG monomer to an NHS-activated Sepharose column (GE Healthcare). Protein was eluted with 3M MgCl$_2$ followed by immediate buffer exchange into Tris-buffered saline pH 8.0 (TBS), and trimers were purified using Superdex 200 16/60 s (GE Healthcare), and then stored in TBS.

V1V2 Fab-Env-8ANC195 complexes were prepared for negative-stain and cryo-EM by incubating purified Env with a 9-fold molar excess of V1V2 Fab (either BG1, PG9, or PG16) overnight at 4°C and then continuing the incubation with a 9-fold molar excess of 8ANC195 Fab. The ternary complex was isolated by SEC using a Superose 6 10/300 column (GE Healthcare).

### X-ray structure determination

Crystals of BG1 Fab were obtained by combining 0.2 μL of a 10 mg/mL protein solution with 0.2 μL of 1.8 M ammonium sulfate, 0.1 M Bis-Tris pH 6.5, 2% v/v PEG 550 MME and 100 nM NDSB-256 at 20°C and cryoprotected in mother liquor supplemented with 20% (v/v) ethylene glycol. X-ray diffraction data were collected using a Pilatus 6M pixel detector (Dectris) at the Stanford Synchrotron Radiation Lightsource beamline 12–2. XDS was used to index, integrate and scale the data (*Kabsch, 2010*), and the structure was refined to 1.9 Å by iteratively refining with Phenix (*Adams et al., 2010*) and manual model building in Coot (*Emsley and Cowtan, 2004*). The BG1 Fab structure was solved by molecular replacement using 2909 Fab (PDB code 3PIQ) V$_H$V$_L$ with CDR loops removed and C$_H$1C$_L$ as search models. The final model (R$_{work}$ = 18.9%, R$_{free}$ = 21.2%) omits the following disordered residues: Ser146$_{HC}$-Gly154$_{HC}$, Gly210$_{HC}$-Thr211$_{HC}$, Lys234$_{HC}$-Thr239$_{HC}$, and Cys214$_{LC}$, and has 96.7%, 3.3% and 0% of residues in the favored, allowed and disallowed regions, respectively, of the Ramachandran plot (*Figure 1—figure supplement 1*).

Hydrogen bonds were assigned using the following criteria: a distance of <3.5 Å, and an A-D-H angle of >90°. Structures were superimposed and figures generated with PyMOL (*Schrödinger, 2011*) and UCSF Chimera (*Pettersen et al., 2004*). Rmsd calculations were done in PyMOL following pairwise Cα alignments without outlier rejection.

### SEC-MALS

Purified Env, Fab, and Env-Fab complexes were characterized by SEC-MALS to determine absolute molecular masses (*Wyatt, 1993*). 80 μL of 1 mg/mL BG505 SOSIP.664 was mixed with a 3-fold molar excess of Fab (PG16, BG1, or 8ANC195$_{G52K5}$) relative to Env trimer, and the reaction volume brought up to 100 μL with sample buffer containing 20 mM Tris (pH 8.0) and 150 mM NaCl. For controls, reactions containing individual components of BG505 SOSIP.664 or Fab were prepared in the same manner. The proteins or protein complexes were then equilibrated overnight at room temperature and injected onto a Superdex 200 10/300 GL gel-filtration chromatography column equilibrated with our sample buffer. The chromatography column was connected with an 18-angle light-scattering detector (DAWN HELEOS II; Wyatt Technology), a dynamic light-scattering detector

(DynaPro Nanostar; Wyatt Technology), and a refractive index detector (Optilab t-rEX; Wyatt Technology). Data were collected at 25°C at a flow rate of 0.5 mL/min every 1 s. The molecular mass of each protein or protein complex was obtained by data analysis using the program ASTRA 6.

## Neutralization assays

Pseudovirus neutralization assays were conducted as described (*Montefiori, 2005*) by the Collaboration for AIDS Vaccine Discovery (CAVD) core neutralization facility for calculation of MNRs (defined in [*Klein and Bjorkman, 2010*]) for BG1 (Figure 2—figure supplement 2). MNRs for other HIV-1 bNAbs came from data published in previous papers (*Galimidi et al., 2015*; *West et al., 2012*). Average IC$_{50}$ values and MNRs are geometric means calculated using the formula ($\prod a_i)^{(1/n)}$; $i$ = 1, 2, ..., $n$. Geometric means are preferred statistics for data sets covering multiple orders of magnitude (*Sheskin, 2004*), as is found for neutralization data across multiple viral strains. For calculating mean IC$_{50}$ values, strains with reported IC$_{50}$s of >$n$ μg/mL were included as $n$ μg/mL. Geometric standard deviations were calculated as the exponential of the standard deviation of the logarithms of the individual MNRs.

## Negative-stain EM data collection and processing

Purified V1V2 Fab-8ANC195-Env complexes (where the V1V2 Fab was either BG1, PG9, or PG16) were diluted to 10 μg/mL in TBS immediately before adding 3 μL to a glow discharged ultrathin C film on holey carbon support film, 400 mesh, Cu grids (Ted Pella, Inc.). Samples were stained with 3% uranyl acetate. Data were collected using FEI Tecnai T12 transmission electron microscope at 120 keV on a Gatan Ultrascan 2k x 2k CCD detector. Images were acquired using a 1 s exposure time at a nominal magnification of 42,000x at 1 μm defocus, resulting in 2.5 Å per pixel. Particles were picked using manual picking in EMAN2.1 (*Tang et al., 2007*), and the CTF correction was done using EMAN2.1 (*Huang et al., 2007*). Initial reference-free 2D class averaging was performed using RELION (*Scheres, 2012*). After 2D classification, particles corresponding to good class averages were selected (*Figure 2—figure supplement 1A–C*), and the particles were further sorted using 3D classification in RELION. A reference model was generated by docking a PG9-V1V2 scaffold structure (PDB 3U4E) onto an Env trimer structure in complex with 8ANC195 (PDB 5CJX) and low pass filtering the model to 60 Å. Refinement for each of the complex structures was conducted using the best reconstruction from the 3D classification as a reference. The resolutions of final reconstructions were calculated with RELION (*Scheres, 2012*) using a gold-standard FSC and a 0.143 cutoff as recommended for resolution estimations for single particle EM reconstructions (*Scheres and Chen, 2012*) (*Figure 2—figure supplement 1D*).

## Cryo-EM data collection and processing

Purified BG1-Env-8ANC195 complex was diluted to 60 μg/mL in TBS and vitrified in liquid ethane using a Mark IV Vitrobot. Sample grids were prepared by adding 3 μL of complex to glow discharged 200 Mesh Quantifoil R1.2/1.3 copper grids. Images were recorded using SerialEM (*Mastronarde, 2005*) on a Titan Krios electron microscope equipped with Gatan K2 Summit direct detector and an energy filter with a slit width of 20 eV. Exposures (15 s) were divided into 50 subframes with a dose rate of 2.92 electrons·pixel$^{-1}$·subframe$^{-1}$. After binning by two and motion correction, each image was 4k ×4k and 1.35 Å per pixel.

The dataset was motion corrected and dose weighted using Unblur and Summovie (*Brilot et al., 2012*; *Campbell et al., 2012*; *Grant and Grigorieff, 2015*). Motion-corrected micrographs without dose weighting were used for contrast transfer function (CTF) estimations. Motion-corrected micrographs with dose weighting were used for particle picking, and motion-corrected micrographs with dose weighting and restored noise power after filtering were used for all classification and refinement processes.

Particles were picked using the SWARM method of EMAN2.1 (*Tang et al., 2007*) (*Figure 3—figure supplement 2A*), and CTF estimations were done using CTFFIND4 (*Rohou and Grigorieff, 2015*) (*Figure 3—figure supplement 2B*). A total of 2837 movies were collected. After motion correction and dose weighting, CTF curves were confidently fit beyond 6 Å in 1371 micrographs; the others were discarded. A total of ~100,000 particles were picked. Particles were classified in 2D with RELION (*Scheres, 2012*), resulting in 1000 2D classes. Of these, 28 classes from 76 k particles were

selected as 'good' classes (*Figure 3—figure supplement 2C*). For 3D classification, the negative-stain BG1-Env-8ANC195 structure (*Figure 2B*) was used as a reference for initial 3D classification; eight 3D classes were then produced. After selecting three 3D classes as 'good' classes, 66 k particles remained for 3D refinement (*Figure 3—figure supplement 2C*). Post-processing, particle polishing, and gold-standard FSC estimations were done after 3D refinement using RELION (*Scheres, 2012*) following procedures in the tutorial. Particle polishing did not improve the resolution of the reconstruction (data not shown), which was estimated to be 6.18 Å (*Figure 3—figure supplement 2A*). Angular distribution plots showing were generated using RELION (*Scheres, 2012*) and visualized in UCSF Chimera (*Goddard et al., 2007*) (Figure 3—figure supplement 3B; *Figure 4—figure supplement 1C*). Local resolution estimations were done using ResMap (*Kucukelbir et al., 2014*) (Figure 3—figure supplement 3C; *Figure 4—figure supplement 1D*).

Using particles in the 6.18 Å reconstruction, we did another round of 3D classification in which the $C_H$-$C_L$ domains of the two BG1 Fabs and the three 8ANC195 Fabs were masked out (*Figure 3—figure supplement 2C*) to account for the ability of Fab $C_H$-$C_L$ domains to adopt multiple conformations with respect to $V_H$-$V_L$ domains due to different elbow bend angles (*Stanfield et al., 2006*). After selecting two 3D classes as 'good' classes, 37k particles remained for 3D refinement ($C_H$-$C_L$ domains were also masked out during 3D refinement), which generated a structure that was also estimated to be 6.18 Å resolution using the gold standard FSC (Figure 3—figure supplement 3A). Unless otherwise noted, the reconstruction derived without masking out $C_H$-$C_L$ domains was used for structural analyses.

Although we used the 2:1 BG1-Env-8ANC195 reconstruction from negative-stain EM as the reference for 3D classification, one subclass (generated from 6.8% of all particles after 2D classification) contained density corresponding to three BG1 Fabs after the first round of 3D classification (*Figure 3—figure supplement 2C*; Figure 3—figure supplement 3D). Particles within that subclass were further refined (using a 60 Å low-pass-filtered reconstruction of that subclass during 3D classification) and post-processed following the standard procedure described in the RELION tutorial, resulting in a 27 Å structure of three BG1 Fabs bound to the Env-8ANC195 complex.

Purified PG9-Env-8ANC195 complexes were used to prepared sample grids as described for BG1-Env-8ANC195 complexes. Images were recorded on a Titan Krios electron microscope equipped with a Falcon II direct detector using FEI's automated data collection software EPU. 2 s exposures were divided into seven subframes, and the dose rate was 13.125 electrons·pixel$^{-1}$·subframe$^{-1}$. After binning by two and motion correction, each image was 4k × 4k and 1.75 Å per pixel. A total of 1767 movies were collected. The dataset was processed as described for the BG1-Env-8ANC195 complex. After motion correction and dose weighting, CTF curves were confidently fit to beyond 8 Å in 1223 micrographs; the others were discarded. A total of ~19k particles were picked. Particles were classified in 2D with RELION (*Scheres, 2012*), resulting in 200 2D classes. Of these, 15 classes from 12k particles were selected as 'good' classes (*Figure 4—figure supplement 1A*). For 3D classification, the negative-stain structure of PG9-Env-8ANC195 (*Figure 2B*) was used as a reference for initial 3D classification; two 3D classes were then produced. After selecting one 3D class as a 'good' class, 9.5k particles remained for 3D refinement (*Figure 4—figure supplement 1A*). The angular distribution of all refinement particles were generated using RELION (*Scheres, 2012*) and visualized in UCSF Chimera (*Goddard et al., 2007*) (*Figure 4—figure supplement 1B*). Local resolution estimations were done by using ResMap (*Kucukelbir et al., 2014*) (*Figure 4—figure supplement 1C*). After 3D refinement, postprocessing and gold-standard FSC estimations were done by using RELION (*Scheres, 2012*) following procedures in the tutorial (*Figure 4—figure supplement 1D*).

## Model building

Coordinates from crystal structures of individual components of the V1V2 Fab-Env-8ANC195 were fit by rigid body docking into cryo-EM density maps. The coordinates of 8ANC195 Fab [Protein Data Bank (PDB 4P9M) and BG1 Fab (this study) were first docked into the corresponding densities, after which three gp41 monomer coordinates from a JR-FL trimer structure (PDB 5FUU) (*Lee et al., 2016*) were fit. We used the JR-FL gp41 coordinates because they included coordinates for the fusion peptide and HR1$_N$ regions (*Lee et al., 2016*). Residues that differed between the JR-FL and BG505 gp41 were altered, and residues outside of EM density were deleted. Three gp120 monomers from a natively-glycosylated BG505 Env structure (PDB 5T3X) (*Gristick et al., 2016*) were fit individually

into protomer densities. Coordinates for ordered N-linked glycans from the natively-glycosylated Env structure (PDB 5T3X) (*Gristick et al., 2016*) were fit separately as rigid bodies at potential N-linked glycosylation sites (PNGSs) at which EM density was apparent (*Figure 3B*). Glycan rings outside of EM density were removed. After initial rigid body docking, refinement of the complex was carried out using phenix.real_space_refine with secondary structure restraints for protein and geometric restraints for protein and N-glycan residues (*Adams et al., 2010*; *Agirre et al., 2015*).

## Accession numbers

CryoEM density maps have been deposited in the Electron Microscopy Data Bank under accession numbers EMD-8693 (BG1-Env-8ANC195 complex) and EMDB-8695 (PG9-Env-8ANC195 complex). Coordinates were deposited in the Protein Data Bank under accession numbers 5VVF (BG1 Fab), 5VIY (BG1-Env-8ANC195 complex), and 5VJ6 (PG9-Env-8ANC195 complex).

## Acknowledgements

We thank Zhiheng Yu, Chuan Hong, and Rick Huang (Janelia Farm) and Mark Yeager and Kelly Dryden (University of Virginia) for assistance with cryo-EM data collection and motion correction, Alasdair McDowall and Songye Chen for training in cryo-EM techniques and data processing, and the Gordon and Betty Moore and Beckman Foundations for gifts to Caltech to support electron microscopy and to the Caltech Molecular Observatory to support X-ray crystallography. Cryo-EM data for the PG9-Env-8ANC195 complex structure was collected at the Molecular Electron Microscopy Core facility at the University of Virginia, which is supported by the School of Medicine and and an NIH grant (G20-RR31199). The Titan Krios microscope and Falcon II direct detector used for data collection at University of Virginia was supported by NIH SIG grant S10-RR025067 and NIH SIG S10-OD018149. Operations at the Stanford Synchrotron Radiation Lightsource are supported by the US Department of Energy and the National Institutes of Health. We also thank Jost Vielmetter and the Caltech Protein Expression Center for transfections and protein expression, and members of the PJB and MCN laboratories for helpful discussions and critical reading of the manuscript. This research was supported by the National Institutes of Health Grant 2 P50 GM082545-06 (to PJB), National Institute Of Allergy and Infectious Diseases of the National Institutes of Health Grant HIVRAD P01 AI100148 (to PJB and MCN), the Bill and Melinda Gates Foundation Collaboration for AIDS Vaccine Discovery Grant 1040753 (to MCN and PJB), and a Comprehensive Antibody-Vaccine Immune Monitoring Consortium Grant 1032144 (MSS). The content is solely the responsibility of the authors and does not necessarily represent the official views of the National Institutes of Health.

## Additional information

### Competing interests

MCN: Senior editor, *eLife*. The other authors declare that no competing interests exist.

### Funding

| Funder | Grant reference number | Author |
| --- | --- | --- |
| Comprehensive Antibody-Vaccine Immune Monitoring Consortium | 1032144 | Michael S Seaman |
| National Institute of Allergy and Infectious Diseases | HIVRAD P01 AI100148 | Michel C Nussenzweig Pamela J Bjorkman |
| Bill and Melinda Gates Foundation | 1040753 | Michel C Nussenzweig Pamela J Bjorkman |
| National Institutes of Health | GM082545-06 | Pamela J Bjorkman |
| National Institutes of Health | G20-RR31199 | Pamela J Bjorkman |
| National Institutes of Health | S10-RR025067 | Pamela J Bjorkman |
| National Institutes of Health | SIG S10-OD018149 | Pamela J Bjorkman |

The funders had no role in study design, data collection and interpretation, or the decision to submit the work for publication.

## Author contributions

HW, Data curation, Formal analysis, Validation, Investigation, Visualization, Writing—original draft, Writing—review and editing, Solved and analyzed EM structures; HBG, Conceptualization, Data curation, Formal analysis, Validation, Investigation, Visualization, Writing—review and editing, Analyzed SEC-MALS experiments; LS, Data curation, Formal analysis, Validation, Investigation, Visualization, Solved the BG1 Fab crystal structure; APW, Conceptualization, Data curation, Formal analysis, Validation, Investigation, Visualization, Writing—review and editing, Performed computational and bioinformatics analyses. Analyzed MNR data; RPG, Data curation, Formal analysis, Investigation, Analyzed MNR data; MSS, Data curation, Conducted in vitro neutralization assays; NTF, Resources, Writing—review and editing, Isolated and characterized the BG1 IgG bNAb; MCN, Conceptualization, Resources, Data curation, Formal analysis, Supervision, Funding acquisition, Investigation, Writing—review and editing, Isolated and characterized the BG1 IgG bNAb; PJB, Conceptualization, Data curation, Formal analysis, Supervision, Funding acquisition, Validation, Investigation, Visualization, Methodology, Writing—original draft, Project administration, Writing—review and editing

## Author ORCIDs

Haoqing Wang, http://orcid.org/0000-0003-0277-3018
Pamela J Bjorkman, http://orcid.org/0000-0002-2277-3990

# Additional files

## Major datasets

The following datasets were generated:

| Author(s) | Year | Dataset title | Dataset URL | Database, license, and accessibility information |
|---|---|---|---|---|
| Haoqing Wang, Pamela J Bjorkman | 2017 | BG1-Env-8ANC195 complex | https://www.ebi.ac.uk/pdbe/emdb/8693 | Publicly available at the EMBL_EBI Protein Dtat Bank in Europe (accession no: EMD-8693) |
| Haoqing Wang, Pamela J Bjorkman | 2017 | PG9-Env-8ANC195 complex | https://www.ebi.ac.uk/pdbe/emdb/8695 | Publicly available at the EMBL_EBI Protein Dtat Bank in Europe (accession no: EMDB-8695) |
| Scharf L, Gristick HB, Bjorkman PJ | 2017 | BG1 Fab coordinate | http://www.rcsb.org/pdb/explore/explore.do?structureId=5VVF | Publicly available at the RCSB Protein Data Bank (accession no: 5VVF) |
| Wang H, Bjorkman PJ | 2017 | BG1-Env-8ANC195 complex coordinates | http://www.rcsb.org/pdb/explore/explore.do?structureId=5VIY | Publicly available at the RCSB Protein Data Bank (accession no: 5VIY) |
| Wang H, Bjorkman PJ | 2017 | PG9-Env-8ANC195 complex coordinates | http://www.rcsb.org/pdb/explore/explore.do?structureId=5VJ6 | Publicly available at the RCSB Protein Data Bank (accession no: 5VJ6) |

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
