## [Decision Letter]

Thank you for submitting your article "Asymmetric recognition of HIV-1 Env trimer by V1V2 loop-targeting antibodies" for consideration by *eLife*. Your article has been reviewed by two peer reviewers, and the evaluation has been overseen by Arup Chakraborty as the Senior Editor and Reviewing Editor. The following individual involved in review of your submission has agreed to reveal his identity: Stephen C. Harrison (Reviewer #1).

The reviewers have discussed the reviews with one another and the Reviewing Editor has drafted this decision to help you prepare a revised submission.

The manuscript characterizes BG-1, an antibody directed at the V1-V2 loop of HIV Env, which is different from other well-studied antibodies that bind this region. In particular, BG-1 has a shorter and less prominent CDRH3 loop than antibodies such as PG9, PG16, and PGT145, and two Fabs (or IgGs) bind per Env, not just one. The manuscript describes (a) a high-resolution crystal structure of the BG-1 Fab, showing a relatively "flat" paratope; (b) a ~6 Å cryoEM structure of a complex of two BG-1 Fabs bound with BG505.664 SOSIP Env; (c) various lower resolution structures, both by negative-stain EM and by cryo, that offer comparison with PG9. The structural work is well described; despite some preferential orientation woes (Figure 3—figure supplement 1), faced by almost everyone in this business, the map in Figure 3 appears to be well interpreted. The manuscript is clear, complete, well-written, soundly interpreted, and well-justified. The experimental scope is exhaustive, leaving "no stone unturned". The results are interesting, with impact for an audience larger than the specializing HIV bNAb community.

We request that the PDB evaluation reports for the reported structures be provided before acceptance. We also request that you consider whether you wish to address the following points prior to publication.

1) A blow-up detail, contoured to show helical rods, should be added to Figure 3—figure supplement 2, as ~6 Å should show nicely defined helices when contoured appropriately. Because preferential orientation will make the gp41 helices along the 3fold axis look particularly nice, the authors should choose other helices, skewed to the preferred direction.

2) There are some issues that might benefit from more detail.

A) Is the binding of these "first" two BG-1 Fabs cooperative, anti-cooperative, or neither? It would be interesting to have a titration curve (from biolayer interferometry?), to add quantitative specifics to the narrative in the fourth paragraph of the Discussion. Clearly, binding of the third ligand is strongly anti-cooperative, for BG505 SOSIP, at any rate. The structure indeed explains why. But for a trimer not stabilized by the SOSIP mutations, which one might expect to fluctuate "open" more readily, would BG-1 then bind with 3:1 rather than 2:1 stoichiometry? Or is the proposed quaternary interaction with a glycan from the neighboring gp120 enough to stabilize the closed state?

B) Related to this last question is the issue of neutralization. How broad is the range of isolates neutralized by BG-1? If Envs that tended to open up readily were actually favored to do so by capture of three BG-1 antibodies, rather than disfavored to do so because of a quaternary interaction of the first antibody to bind, then one might even imagine the apparently paradoxical outcome that these antibodies would neutralize tier 2 strains better than tier 1s (since tiers 1 and 2 correlate with stability of the "closed" state with respect to "open" ones).

Perhaps the best way to summarize both these points is to ask: how "quaternary" is BG-1? The structure may not answer this question definitively (limitation of resolution, fluctuation of glycan, etc.), but binding data might shed light on it. The broader point has to do with the desirability of trying to elicit antibodies of this kind with a vaccine. The paper would be even stronger if this point could be argued directly. Perhaps the relevant data are available but not included here. If so, it would help at least to summarize them. The manuscript is suitable for *eLife*, precisely because it illustrates these general but also subtle issues, appreciated by very few in the HIV field. The authors do appreciate them (as far as this reader can tell), but it would therefore be good for them to be more explicit.

---

## [Author Response]

*[…] We request that the PDB evaluation reports for the reported structures be provided before acceptance. We also request that you consider whether you wish to address the following points prior to publication.*

Submitted with the revised manuscript.

*1) A blow-up detail, contoured to show helical rods, should be added to Figure 3—figure supplement 2, as ~6 Å should show nicely defined helices when contoured appropriately. Because preferential orientation will make the gp41 helices along the 3fold axis look particularly nice, the authors should choose other helices, skewed to the preferred direction.*

Panel D in Figure 3—figure supplement 2 now shows a density map in which helices can be seen in a variety of orientations. In addition, we submitted a supplemental video with the revised manuscript that shows the model and density while rotating, thus densities for all parts of the model can be examined.

*2) There are some issues that might benefit from more detail.*

*A) Is the binding of these "first" two BG-1 Fabs cooperative, anti-cooperative, or neither? It would be interesting to have a titration curve (from biolayer interferometry?), to add quantitative specifics to the narrative in the fourth paragraph of the Discussion. Clearly, binding of the third ligand is strongly anti-cooperative, for BG505 SOSIP, at any rate. The structure indeed explains why. But for a trimer not stabilized by the SOSIP mutations, which one might expect to fluctuate "open" more readily, would BG-1 then bind with 3:1 rather than 2:1 stoichiometry? Or is the proposed quaternary interaction with a glycan from the neighboring gp120 enough to stabilize the closed state?*

These are very interesting questions. We’re not sure how to use biolayer interferometry to gain information about whether the BG1 binding interaction is cooperative or not (also, we have no access to a BLI Octet machine). However, we routinely perform binding studies by surface plasmon resonance (SPR using a Biacore), an alternative technology for real-time label-free detection of binding interactions. In this case, we have never observed evidence of cooperative binding for Fabs interacting with Env trimer. It is certainly possible to model binding data using various models postulating conformational change(s) and/or cooperativity, but this is discouraged by Rich and Myszka in their 2009 review (Grading the commercial optical biosensor literature – Class of 2008: ‘The Mighty Binders’) because data (of any sort) will always be better fit by adding extra parameters to a model. Cooperativity or anti-cooperativity would be reflected in the Hill coefficients of equilibrium binding curves (or in vitro neutralization curves).

Unfortunately, other factors, such as heterogeneity, also influence observed Hill coefficients. Indeed, gp120 glycan heterogeneity is likely to have a substantial effect on such curves for BG1, and it would be difficult or impossible to deconvolute the effects of cooperativity and heterogeneity. Thus, we do not believe that we can accurately measure the degree of cooperativity of multiple BG1 Fabs binding to Env.

*B) Related to this last question is the issue of neutralization. How broad is the range of isolates neutralized by BG-1? If Envs that tended to open up readily were actually favored to do so by capture of three BG-1 antibodies, rather than disfavored to do so because of a quaternary interaction of the first antibody to bind, then one might even imagine the apparently paradoxical outcome that these antibodies would neutralize tier 2 strains better than tier 1s (since tiers 1 and 2 correlate with stability of the "closed" state with respect to "open" ones).*

This is also a very interesting issue. Thanks for raising it.

Regarding the breadth of BG1: it is not as broad as PG9 or PG16, largely because of its sensitivity to glycosylation at position 130 (as discussed in the Comparison of BG1 and PG9 epitopes and paratopessection) (see Freund et al., 2017, for neutralization data). Comparing the activities of BG1 and PG9 on tier 1 strains vs. non-tier 1 strains, both antibodies show somewhat greater potency and breadth on tier 1 strains (see Figure 8 – coverage curves figure). Thus, BG1 does not neutralize tier 2 strains better than tier 1 strains. Neutralization studies (such as Brandenbert et al., PLoS Pathog. 13, e1006313, 2017) indicate that a single antibody molecule binding to an Env trimer is generally sufficient to neutralize that trimer, and since a BG1 IgG molecule can only bind with a single Fab at a time, strains able to bind three BG1s per spike might not have greater BG1 sensitivity. Hence, if tier 1 strains have more open Env trimers that can bind three BG1 IgGs, this might not be apparent in the BG1 IC50s of those strains.

Author response image 1.**DOI:**
http://dx.doi.org/10.7554/eLife.27389.020

*Perhaps the best way to summarize both these points is to ask: how "quaternary" is BG-1? The structure may not answer this question definitively (limitation of resolution, fluctuation of glycan, etc.), but binding data might shed light on it. The broader point has to do with the desirability of trying to elicit antibodies of this kind with a vaccine. The paper would be even stronger if this point could be argued directly. Perhaps the relevant data are available but not included here. If so, it would help at least to summarize them. The manuscript is suitable for eLife, precisely because it illustrates these general but also subtle issues, appreciated by very few in the HIV field. The authors do appreciate them (as far as this reader can tell), but it would therefore be good for them to be more explicit.*

Given that BG1 is more heavily somatically mutated than more potent V1V2 bNAbs such as PG9 and PG16, we think it is probably not desirable to try to deliberately raise this type of antibody by vaccination. However, it is likely that immunogens designed to target the V1V2 site would raise both PG9- and BG1-types of antibodies. As discussed at the end of the Discussion section of the paper, immunogens designed to raise bNAbs against the V1V2 epitope would require removal of the Asn130 glycan if the goal was to target both types of antibodies. Keeping the Asn130 glycan would likely discourage BG1-style antibodies, but would not be incompatible with PG9-style antibodies.